# Field Application of a New CSF Vaccine Based on Plant-Produced Recombinant E2 Marker Proteins on Pigs in Areas with Two Different Control Strategies

**DOI:** 10.3390/vaccines9060537

**Published:** 2021-05-21

**Authors:** Yeonsu Oh, Youngmin Park, Bo-Hwa Choi, Soohong Park, Sungmin Gu, Jungae Park, Jong-Kook Kim, Eun-Ju Sohn

**Affiliations:** 1College of Veterinary Medicine and Institute of Veterinary Science, Kangwon National University, Chuncheon 24341, Korea; yeonoh@kangwon.ac.kr; 2BioApplications Inc., Pohang Techno Park Complex, 394 Jigok-ro, Pohang 37668, Korea; ympark86@postech.ac.kr (Y.P.); bhchoi@bioapp.co.kr (B.-H.C.); saysh13@bioapp.co.kr (S.P.); gusungmin@bioapp.co.kr (S.G.); oh@bioapp.co.kr (J.P.); bio2020@bioapp.co.kr (J.-K.K.)

**Keywords:** classical swine fever virus, E2 glycoprotein, DIVA vaccine, plant-made vaccine, field application

## Abstract

A classical swine fever virus (CSFV)-modified live LOM (low-virulence strain of Miyagi) vaccine (MLV-LOM) to combat CSF has been used in places where the disease is prevalent around the world, including in Korea, except in Jeju Island. In general, modified live virus-based vaccines (MLV) are known to be highly effective in inducing immune responses. At the same time, MLVs also have potential dangers such as a circulation in the field. There is still a need for safer and more effective vaccines to control CSF in the field. In this study, we applied a new CSF vaccine based on plant-produced recombinant E2 marker proteins at two different locations, Jeju Island and a suburb of Pohang, using different CSF control strategies. The result suggested that vaccinated sows in Jeju Island highly developed immunogenicity and maintained stably until 102 days post-vaccination (dpv). Its piglets that received maternal antibodies were shown to carry high serological values and maintained them until 40 days of age, which was the end of the follow-up. Naïve piglets vaccinated at 40 days of age showed high serological values and these were maintained until 100 days of age (60 dpv), which was the end of the follow-up. The vaccine was also effective in inducing immune responses in newborn piglets that carried maternal antibodies received from MLV-LOM vaccine-immunized mother sows.

## 1. Introduction

Classical swine fever virus (CSFV) is a deadly virus for pigs and is registered as a Class I legal infectious disease of animals in Korea, and listed as a notifiable disease by the World Organization for Animal Health (OIE) [1]. It is highly contagious and causes multisystemic pathogenicity in pigs, resulting in serious economic loss. In Korea, the government has maintained a policy to use a CSF-modified live LOM (low virulence strain of Miyagi) vaccine (MLV-LOM) in pigs since 1974, although adverse effects have been intermittently raised, because the benefits of carrying on with vaccination as normal among pigs are far greater than the risk of extra deaths from CSFV. Meanwhile, the quarantine authorities have been carrying out a project, “CSF-free” nationwide, starting with Jeju Island which is geographically isolated from the mainland of Korea [2,3]. As part of this project, the MLV-LOM vaccine has been discontinued in Jeju Island since 2000. Recently, various adverse effects including abortion and stillbirth in pregnant sows were reported in the mainland of Korea, being related to using the MLV-LOM vaccine. In addition, similar clinical manifestations in Jeju Island have been described [4]. Subsequently isolated CSFVs from Jeju pigs were found to be similar to LOM strains (Jeju LOM strains), likely derived from the MLV-LOM vaccine. Furthermore, these LOM strains were transmitted from pregnant sows to live offspring, resulting in wide spread of the virus. Another study showed that CSFV field isolates on Jeju Island were most closely related to the LOM strain, but appeared to have undergone substantial changes throughout the entire genome, particularly in the NS2 and 3′-untranslated regions [5]. Moreover, based on the case study carried out on Jeju Island, Korea, these authors noted that the possibility of genetic drift at other hotspots in the process of viral fitness selection cannot be excluded in the future [5]. Likewise, these studies warn the dangers of using live viral strains for pig vaccination.

A recent advance in vaccine development introduced subunit vaccines, which are based on recombinant proteins coded by viral genes [6]. Three different glycoproteins (E1, E2 and E^rns^) are presented on the envelop of CSFV, and the E2 protein is capable of inducing the production of neutralizing antibodies [7,8]. Since the subunit vaccine using E2 as an antigen does not have E^rns^, it is possible to distinguish serologically whether animals have been vaccinated or infected with CSFV through the presence or absence of antibody formation against E^rns^. Recombinant protein-based vaccines can eliminate all possible dangers derived from viruses used as vaccines, such as the persistent survival of LOM strains in the field after the strains were introduced into clean areas like Jeju Island where a non-MLV-vaccine policy has been adhered to maintain a CSF-free region [4] and the possible risk of reversion to virulent forms. These vaccines also serve as DIVA vaccines, because they allow differentiation of infected from vaccinated animals (DIVA). Although they have many advantages over conventional virus-based vaccines, it is technically challenging to design highly effective recombinant protein vaccines, and also to produce them at an affordable price for livestock. Conventionally, recombinant proteins are largely produced in bacteria, insect, and/or animal cells, which are based on large fermenters [9]. Recently, plants were introduced as a more affordable system for producing recombinant proteins [10,11]. In fact, plants can be a rapid as well as safe system for the large-scale production of vaccines especially in urgent situations, such as pandemic outbreaks or bioterrorism [12]. Currently, plant systems are being actively pursued to produce recombinant protein vaccines for livestock [13]. The first approved vaccine was for the Newcastle disease virus of chicken, which was produced in BY-2 cells [14]. However, it could not be introduced into a marketplace due to low expression levels of the recombinant protein resulting in high prices. Since then, there was a great deal of progress in improving expression levels and also the purification of recombinant proteins from plant extracts.

We recently developed a subunit vaccine based on recombinant E2 proteins of CSFV produced in *Nicotiana benthamiana* as a green marker vaccine [15]. It was approved by the Korean government under the name HERBAVAC^TM^ as a commercial CSF vaccine for swine industry in Korea. However, before its introduction into the market, it was necessary to test its efficacy in the field, where animal conditions vary greatly depending on individual farms. This is the report in the line of examining immune induction of antibodies in response to HERBAVAC^TM^ under field conditions with and without pre-existing CSF antibodies. We performed field trials at two different locations, Jeju Island and the suburb of Pohang in the mainland of Korea. These two locations provided animals with two different conditions: Pigs on farms in the suburb of Pohang have been subjected to mandatory vaccination against CSF, and thus even newborn piglets should carry maternal E2 antibodies received from mother sows, whereas pigs on farms on Jeju Island are naïve to CSFV without prior exposure to any CSF vaccines.

## 2. Materials and Methods

### 2.1. E2 Recombinant Protein Production, Purification, and Vaccine Formulations

Development of a plant-produced E2 recombinant protein as a CSFV subunit marker vaccine was described in a previous study [15]. Briefly, T4-generation transgenic plants (*N. benthamiana*) harboring the chimeric E2 recombinant gene were grown hydroponically in a growth facility for 4 weeks. Aerial parts of plants were harvested and used for purification of the E2 recombinant protein as E2 antigen. Large-scale production of the E2 antigen was carried out under conditions from the Good Manufacturing Practice for Veterinary Pharmaceutical in Korea (KVGMP). Purified E2 antigen was mixed with the adjuvant Emulsigen^®^-D (MVP Adjuvant^®^, Omaha, NE, USA). The concentration of E2 antigen in the vaccine formula was 100 µg/mL. In manufacturing the vaccine, we complied with the national shipping standards, and no abnormalities were found in the safety tests in test animals and target animals.

### 2.2. Animals and Vaccination Trials

#### 2.2.1. Jeju Island

Nine pregnant sows from three different farms (named Jeju A, B, and C) were purchased, which had not been exposed to CSFV or any CSF vaccine were secured. Sera were obtained from these animals 70 days before delivery and tested for the presence of anti-E^rns^, anti-E2, and neutralizing antibodies (NA). Those pregnant sows were vaccinated three times intramuscularly on the neck with 2 mL of HERBAVAC^TM^ at 70, 49, and 28 days antepartum (0, 21, and 42 days post-vaccination; dpv) (Figure 1a). Sera were collected just before each vaccination (at 0, 21, and 42 dpv) and at 20 and 40 days postpartum (62 and 102 dpv), and tested for E2-specific antibodies and NA. Clinical symptoms of sows were monitored daily. Newborn piglets from those vaccinated sows were randomly allocated and collected with sera to examine the level of maternally derived E2-specific antibodies and NA at 20 and 40 days after birth (Figure 1a).

Together, to examine immunogenicity in naïve young piglets that had not been exposed to CSFV or any CSF vaccine, 12 piglets from each farm were randomly selected and injected intramuscularly twice on the neck with 1 mL of the vaccine at the age of 40 and 60 days after birth (0 and 20 dpv). Clinical symptoms of piglets were monitored daily. Sera were collected and tested for E2 antigen-specific immune responses at the age of 40 (before vaccination, 0 dpv), 60 (20 dpv), and 100 (60 dpv) days (Figure 1b).

#### 2.2.2. Suburb of Pohang

Twenty piglets were randomly selected from three different farms (named Pohang A, B, and C). Animals were injected intramuscularly twice on the neck with 1 mL of the vaccine at the age of 40 and 61 days (0 and 21 dpv) (Figure 2). Clinical symptoms of piglets were monitored daily. Sera were collected and tested for E^rns^-specific (at only 33 days of age), E2-specific antibodies, and NA at the age of 33 (7 days before vaccination, 0 dpv), 54 (14 dpv), 81 (41 dpv), and 137 (97 dpv) days, respectively.

### 2.3. Detection of E^rns^-Specific and E2-Specific Antibodies

A commercial ELISA kit, pig-type CSFV Erns Ab (INDICAL BIOSCIENCE, Leipzig, Germany), was used to measure levels of E^rns^-specific antibodies, in accordance with the manufacturer’s instruction. An S/P (sample compared with the positive control) value less than 0.3 was considered to be negative, and an S/P value equal to or greater than 0.5 was considered to be positive. A commercial BioNote CSFV Antibody B-ELISA kit (BioNote, Hwaseong, Korea) was used to measure levels of E2-specific antibodies, in accordance with the manufacturer’s instruction. A percent inhibition value equal to or greater than 40 was considered to be positive, and a value less than 40 was considered to be negative.

### 2.4. Neutralizing Antibodies

To obtain the titer of NAs, serum samples were subjected to a serial twofold dilution, and examined by the neutralization peroxidase-linked assay. Titers were expressed as the reciprocal fold dilution of sera that neutralized 100 × tissue culture infective dose (TCID)_50_ of the CSFV LOM strain in 50% of culture replicates [16].

### 2.5. Statistical Analysis

Summary statistics were calculated for all groups to assess the overall quality of the data, including normality. The data were analyzed by one-way analysis of variance (ANOVA), which was followed by Tukey’s multiple-comparison test for pairwise testing. A linear regression was performed to determine the correlation between antibody titers before and after vaccination, and its increment (delta value) at days post-vaccination. The delta value was defined as the value of antibody titers between the time before vaccination and the testing time points. Numerical serology values were expressed as mean ± standard deviation (SD).

## 3. Results

### 3.1. Field Application at Farms on Jeju Island with Naïve Pregnant Sows and Young Piglets

Pregnant sows in this experimental setting did not have any prior exposure to CSFV or CSF vaccines according to the CSF control measure in Jeju Island. No suboptimal clinical signs were not observed at the time of vaccination or during experiments. Shortly after starting the experiment, one sow in the ninth pregnancy at Jeju A farm was dead due to fracture and lameness of its hind legs.

Serology analyzed for three criteria, E^rns^-, E2-specific antibodies, and NA titers at 70 days antepartum (before primary injection of HERBAVAC^TM^), was all negative for both the E^rns^-specific antibody ELISA and NA titers, and showed negligible levels in the E2-specific antibody ELISA titer except for one animal. It was later confirmed that they were not exposed to any vaccine or natural infection (Table 1). The one outlier for the E2-specific antibody ELISA was possibly caused by a non-specific reaction of the ELISA test used. At each time before vaccination and at 20 and 40 days postpartum, serology was tested for all sows. At 21 dpv, immediately before the 2nd vaccination, the E2-specific antibody ELISA titer was observed in all tested sows except one. The E^rns^-specific antibody ELISA and NA titers were all negative except one, which was positive for both the E2-specific antibody ELISA and the NA titers (Table 1). Serology before the third vaccination (at 28 days antepartum and 42 dpv) was all positive for both the E2-specific antibody ELISA and NA titers. The E2-specific antibody ELISA values ranged from 83.93 to 94.43 with a mean value of 87.78 ± 1.07 (SD, *n* = 2) at Jeju A, 89.97 ± 4.85 (SD, *n* = 3) at Jeju B, and 92.18 ± 0.41 (SD, *n* = 3) at Jeju C, thereby greatly exceeding 40, the criterion for the positive response (Table 1). NA titers ranged from 1:512 to 1:2048 at Jeju A, from 1:512 to 1:1024 at Jeju B, and from 1:1024 to 1:2048 at Jeju C, again greatly exceeding the positive criterion, 1:32. At 62 dpv (20 days postpartum) seven sows were positive for both the E2-specific antibody ELISA and NA titers without much change from those at 42 dpv. At 102 dpv (40 days postpartum) eight sows still had high values for both the E2-specific antibody ELISA and NA titers, but both values tended to decrease slightly compared with those at 62 dpv (Table 1).

Serology of newborn piglets from vaccinated sows with HERBAVAC^TM^ showed very high E2-specific antibody ELISA titers among 20-day-old piglets, ranging from 92.01 to 96.58 with a mean value of 93.22 ± 0.74 (SD, *n* = 3) at Jeju A, 96.27 ± 0.38 (SD, *n* = 4) at Jeju B, and 94.20 ± 2.17 (SD, *n* = 3) at Jeju C. NA titers were 1:1024 at Jeju A, and ranged from 1:1024 to 1:4096 at Jeju B, and from 1:512 to 1:4096 at Jeju C (Table 2). Next, to investigate the maintenance of antibodies in piglets, sera obtained from 40-day-old piglets were analyzed with the E2-specific antibody ELISA and NA titers. Both E2-specific antibody ELISA and NA titers showed only a marginal decrease 40 days after birth. (Table 2).

Next, separate from the sow vaccination test, 36 piglets were randomly selected from three different farms (12 animals from each farm), and vaccinated twice with HERBAVAC^TM^ at 40 and 60 days after birth (0 and 20 dpv). Serology was analyzed on days 40, 60, and 100 after birth (0, 20, and 60 dpv) for the E^rns^- and E2-specific antibody ELISA, and NA titers. Subsequently, 34 animals (11, 11, and 12 animals from Jeju A, Jeju B, and Jeju C, respectively) before immunization were negative for the E^rns^-specific antibodies, indicating that the corresponding animals were not exposed to CSFV or the CSF vaccine (data- not-shown). At 20 dpv (60 days of age), the majority of them (24 of 31) tested positive for the E2-specific antibody ELISA (Figure 3a), and a half of them (18 of 36) were positive for the NA titer (Figure 3b). At 60 dpv (100 days of age), 33 out of 34 pigs were positive for the E2-specific antibody ELISA and NA titers. Antibody ELISA values ranged from 73.0 to 96.45 with a mean value of 94.79 ± 1.69 (SD, *n* = 10) at Jeju A, 81.07 ± 27.39 (SD, *n* = 12) at Jeju B, and 91.87 ± 1.98 (SD, *n* = 12) at Jeju C. NA titers ranged from 1:512 to 1:2048 at Jeju A, from 1:256 to 1:2048 at Jeju B, and from 1:512 to 1:2048 at Jeju C. The serological increment was shown to be statistically significant in all three farms (*p* < 0.05).

### 3.2. Field Application with Piglets with Maternal E2 Antibodies at Farms in the Suburb of Pohang

In this experimental setting, we examined whether HERBAVAC^TM^ could induce antibody production in young piglets that had maternal E2 antibodies received from vaccinated mother sows. Twenty 33-day-old piglets were randomly selected by weight and sex from three farms in the suburb of Pohang (Pohang A, B, and C). These piglets were born from mother sows that had been immunized with the MLV-LOM strain. Thus, these young piglets should have carried maternal antibodies at varying degrees depending on the individual animal.

The antibody E^rns^ ELISA titer varied from farm to farm; in particular, those at Pohang B were somewhat higher than those of other farms at 7 days before vaccination (Figure 4a). The E^rns^-specific antibodies, however, decreased at 14 dpv. The antibody E2 ELISA titers appeared in various range among all animals of all tested farms at 7 days before vaccination, and a slight increase was found in pigs of Pohang A at 14 dpv, but the titer decreased in pigs of the other farms (Figure 4b). At 41 dpv (81 days of age), pigs from all tested farms presented fully increased antibody E2 ELISA titers and maintained these until 97 dpv (137 days of age).

The NA titers were found in all tested pigs before vaccination, slightly decreased at 14 dpv, and highly increased at 41 dpv (Figure 4c). These elevated NA titers were maintained until the last day of follow-up, at 97 dpv. The serological increment from E2 antibody ELISA and NA titers was shown to be statistically significant in all three farms at 41 and 97 dpv (*p* < 0.05).

## 4. Discussion

It has been discussed for decades how to monitor and control virulent animal epidemics, and the know-how that has resulted from these efforts might give some insights on dealing with large-scale pandemic diseases in human [17]. Biotechnology has been dedicated to developing safer and more effective recombinant protein subunit vaccines, such as porcine circovirus 2 (PCV2) or porcine parvovirus (PPV) virus-like particle (VLP) vaccines, consisting of coat protein only without nucleic acid [18,19].

The present study was conducted (1) to evaluate the immunogenicity of piglets born to sows vaccinated with the green marker vaccine, HERBAVAC^TM^ produced in plants, where CSF has never been experienced as either a vaccine or a virus, (2) to evaluate the immunogenicity of piglets vaccinated with the HERBAVAC^TM^, where CSF has never been experienced, and (3) to evaluate the immunogenicity of HERBAVAC^TM^-vaccinated piglets born to sows vaccinated with a CSF vaccine, where CSF has been commonly experienced as a vaccine or as a virus.

Vaccinated sows maintained high levels of CSFV E2-specific antibody ELISA and NA titers at 102 dpv (40 days after delivery). The results suggested that positive immune responses were maintained 60 days after the third immunization. Piglets born to the vaccinated sows showed very high values of E2-specific antibody ELISA and NA titers, indicating a very efficient transfer of maternal antibodies, and the raised immunogenicity was conserved with a marginal decrease 40 days after birth. The serology data of piglets appeared to be higher than those in pregnant sows. It suggested that maternally derived antibodies were safely uptaken by newborn piglets and successfully developed as their own immunogenicity.

Sow vaccination has several advantages: Not only does it protect young piglets from various pathogens during vulnerable periods in the early stages of life, but it also reduces the cost of piglet vaccination in some cases. Vaccination of sows protected the piglets against viral challenge up to 8 weeks of age [20]. Sow and piglet vaccination regimens can control PCV2 infection from weaning to early weaned periods by the passive transfer of maternal antibodies from vaccinated sows and from late weaned to fininshing by active immunogenicity developed by piglet vaccination [21].

Naïve piglets in Jeju with no vaccination policy against CSFV were vaccinated with HERBAVAC^TM^, and developed and maintained high values of antibody titers stably at least until the end of experiment (100 days of age; 60 dpv) in the field. Piglets in Pohang with a vaccination policy against CSFV carried antibodies before the vaccination was transferred from vaccinated sows. The vaccinated piglets with HERBAVAC^TM^ formed high antibody titers, either CSFV E2 antibody ELISA or NA titers, and maintained them until the end of experiment, at 97 dpv, that is, 137 days of age.

In areas where sows are vaccinated, interference with maternally transferred antibodies in piglet vaccination has always been a concern in fields. However, with the combination of sow and pig vaccinations, the piglet vaccinations were effective at protecting pigs against PCV2 infection until the age of 25 weeks, whether the piglet vaccination was performed at weaning, 21 days of age, or delayed until 49 days of age [21].

## 5. Conclusions

Here we confirmed that the commercialized plant-produced E2 subunit vaccine is highly efficient in generating CSFV E2-specific antibodies in both piglets born to MLV-LOM-vaccinated sows at three different farms in Pohang and piglets E2-specific antibody-free at three different farms in Jeju. This is the first report evaluating a licensed plant-produced swine recombinant subunit vaccine at multiple pig farms. Moreover, animals at the farms in Jeju were not vaccinated for PCV2 and porcine reproductive and respiratory syndrome (PRRS), whereas animals at the farms in Pohang were subjected to vaccination for PCV2 and PRRS, indicating that HERBAVAC^TM^ and other types of vaccines do not have any cross-interference. In the case of Pohang, the level of E^rns^ antibody was an important parameter to determine the maintenance period of maternal antibodies by live vaccines. The result showed that the E^rns^ antibody level changed from positive to negative around 54 days of age. Since the first injection was performed at 40 days of age, it suggested that the effect of the marker vaccine was expressed well without problems even in E^rns^-positive pigs. However, we note that future studies should be carried out to investigate the sustainability of antibodies, the half-life of antibodies, maternally transferred antibody titers in newborns, and vaccine breakage in piglets. In addition, analysis of animal productivity would help to assess the benefits of recombinant subunit vaccines.

## Figures and Tables

**Figure 1 vaccines-09-00537-f001:**
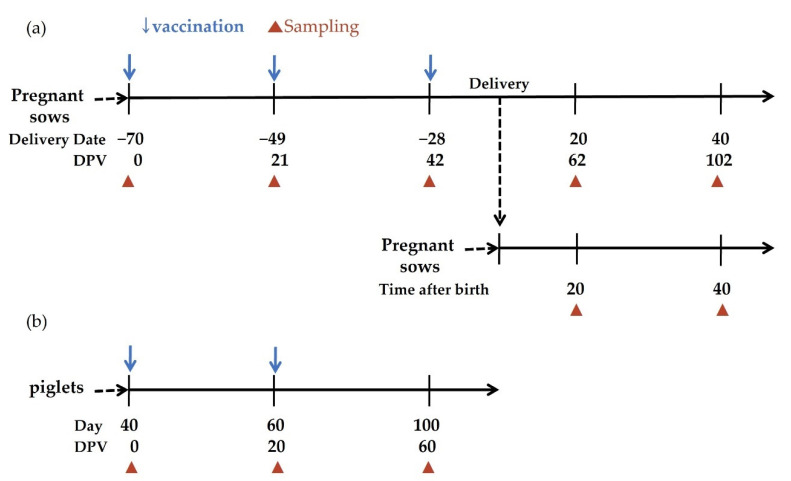
The schematic presentation of Immunization protocols in Jeju Island. (**a**) The timing of blood collection, vaccination, and delivery from pregnant sows and newborn piglets was presented. (**b**) The timing of blood collection and vaccination from CSF-free piglets was presented.

**Figure 2 vaccines-09-00537-f002:**
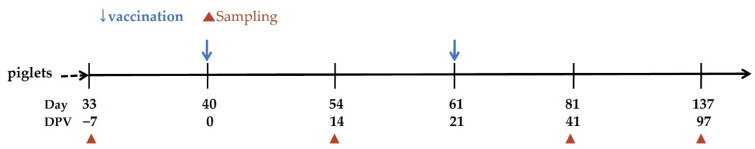
The schematic presentation of immunization protocol in the suburb of Pohang. The timing of blood collection and sampling from the piglets was presented.

**Figure 3 vaccines-09-00537-f003:**
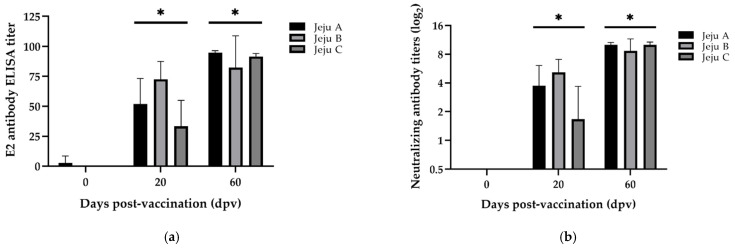
Serological analysis of serum samples from piglets on farms on Jeju Island. E2-specific ELISA (**a**) and neutralizing antibody titers (**b**) from the sera collected at 0, 20, and 60 dpv are presented. Bars represent the mean ± standard deviation (SD) during animal experiment period. * *p* < 0.05.

**Figure 4 vaccines-09-00537-f004:**
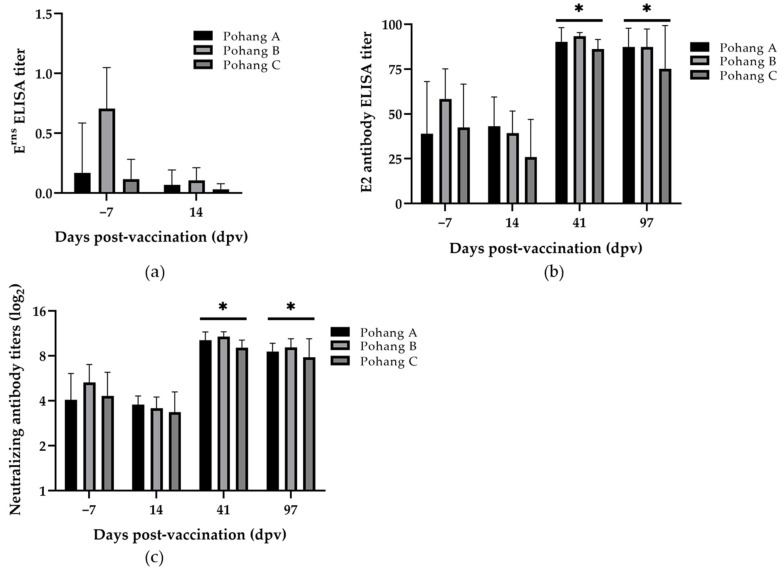
Serological analysis of serum samples from piglets on farms in Pohang. Erns-specific ELISA (**a**), E2-specific ELISA (**b**), and neutralizing antibody titers (**c**) were presented from the sera collected at different time points. Bars represent the mean ± standard deviation (SD) during animal experiment period. * *p* < 0.05.

**Table 1 vaccines-09-00537-t001:** Serological analysis of serum samples from pregnant sows on farms on Jeju Island.

Delivery Date		−70			−49			−28		20		40	
Day Post-Vaccination (dpv)		0			21			42		62		102	
		1st Vaccination			2nd Vaccination			3rd Vaccination		-		-	
Test		E^rns 1^	E2 ^2^	NA ^3^	E^rns^	E2	NA	E2	NA	E2	NA	E2	NA
Jeju A	1	0.00	2.35	- ^4^	0.00	42.56	1/8	90.71	1/2048	89.21	1/512	83.40	1/128
	2	0.00	3.84	-	0.00	30.38	-	88.85	1/512	73.14	1/128	60.51	1/32
	3	0.00	0.00	-	* D	* D	* D	* D	* D	* D	* D	* D	* D
Jeju B	1	0.00	41.13	-	0.00	50.23	1/16	83.93	1/512	80.51	1/256	54.40	1/32
	2	0.00	0.00	-	0.00	45.37	1/8	91.56	1/1024	92.96	1/1024	89.81	1/128
	3	0.00	1.38	-	0.00	47.72	1/8	94.43	1/1024	94.36	1/512	74.99	1/64
Jeju C	1	0.04	0.00	-	0.00	67.77	1/32	91.82	1/1024	94.10	1/1024	93.80	1/512
	2	0.00	11.10	-	** N/A	** N/A	** N/A	92.01	1/2048	94.30	1/1024	92.20	1/256
	3	0.00	0.26	-	** N/A	** N/A	** N/A	92.70	1/1024	** N/A	** N/A	94.65	1/512

^1^ E^rns^ S/P (sample compared with the positive control): Positive, S/P ≥ 0.5; suspect, 0.3 ≤ S/P < 0.5; negative, S/P < 0.3. ^2^ E2 PI (percent inhibition): Positive, PI ≥ 40; negative, PI < 40. ^3^ NA (neutralizing antibody): Positive, NA ≥ 1/32; negative, NA < 1/32. ^4^ Negative values are replaced with zero. * D, died. ** N/A, not available.

**Table 2 vaccines-09-00537-t002:** Serological analysis of serum samples from newborn piglets on farms on Jeju Island.

Time after Birth (days)		20		40	
Test		E2 ^1^	NA ^2^	E2	NA
Jeju A	1	92.73	1/1024	87.36	1/256
	2	94.07	1/1024	87.30	1/128
	3	92.86	1/1024	84.82	1/128
Jeju B	1	96.58	1/1024	93.21	1/512
	2	96.12	1/2048	90.17	1/1024
	3	95.80	1/1024	93.44	1/512
	4	96.58	1/4096	94.36	1/2048
	5	** N/A	** N/A	92.32	1/1024
Jeju C	1	96.35	1/1024	95.27	1/1024
	2	92.01	1/1024	96.43	1/1024
	3	94.23	1/1024	94.06	1/1024
	4	** N/A	1/4096	95.25	1/4096
	5	** N/A	1/2048	96.72	1/2048
	6	** N/A	1/512	84.44	1/512
	7	** N/A	** N/A	96.28	1/2048
	8	** N/A	** N/A	94.86	1/4096

^1^ E2 PI (percent inhibition): Positive, PI ≥ 40; negative, PI < 40. ^2^ NA (neutralizing antibody): Positive, NA ≥ 1/32; negative, NA < 1/32. ** N/A, not available.

## Data Availability

The data presented in this study are available on request from the corresponding author.

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
