# Peer review of "Field Application of a New CSF Vaccine Based on Plant-Produced Recombinant E2 Marker Proteins on Pigs in Areas with Two Different Control Strategies"

_vaccines, 2021, doi:10.3390/vaccines9060537_

Round 1

Reviewer 1 Report

In this study, the authors perform a follow-up to their previous work on a tobacco plant-grown subunit vaccine against Classical swine fever virus.  The work is well done, and nicely presented, with small but reasonable numbers of experimental animals in each group.

The only substantial problem that I see with the report is in the explanation of the mechanism of antibody transfer. 

Line 183 – the phrase “during breastfeeding”

As far as I am aware, antibodies delivered into a piglet via colostrum will tend to be IgA, and will tend to remain within the enteric tract, much as would be the case in a human infant.  Yet the amount of total and neutralizing antibody detected in the serum of the piglets 20 days after delivery is still roughly equivalent to that in the sows 28 days before delivery.  As I understand it, the Bionote E2 antibody ELISA being used for detection is based on the principle of antibody blocking rather than antibody capture, so it would not be able differentiate between isotypes of immunoglobulin, meaning there would not be a convenient way to tell presumably non-maternal IgG from presumably maternal IgA, as well as being no clear mechanism to explain how either of the antibodies would cross from the gut to the bloodstream in the quantities observed, via colostrum.  As it says on line 191 – the antibody levels are higher in the piglets than the sows in some cases, which might be technically possible but strikes me as rather unlikely, given the rapid decline in serum antibody levels in vaccinated sows, contrasted with the relatively constant serum antibody levels in piglets that were exposed to the vaccine in utero. 

I think it is a reasonable possibility, not excluded by the experiments in this study, that the immunoglobulin levels may be driven by circulating B cells in the piglets.  Testing this hypothesis, as well as follow-up to determine whether those are of piglet or maternal origin is probably outside the scope of the present study.  To justify this line of reasoning, that piglets pick up antibody via colostrum rather than via immunization in utero, please perform an experiment in which piglets from unvaccinated sows are allowed to feed from vaccinated sows and examine the antibody level as well as antibody isotype in the bloodstream.  Alternatively, please re-word the manuscript so that it does not imply maternal transfer of antibodies direct to the bloodstream over a month after delivery when maternal antibody levels are waning.    

Author Response

Dear Reviewer,

We attached responses to your comments.

Reviewer 2 Report

This work is very interesting to reades and facilitate the wide range of applications of HERBAVACTM vaccine agaisnt CSFV infection in the future. While it's better to show the results with figures not just table. The discusion is very concise and can be approvedFunding should be differnt to  Acknowledgment.

Author Response

Dear reviewer,

We attached responses to your comments.

Reviewer 3 Report

Summary: This study seeks to evaluate seroconversion in pigs following administration of a plant-produced recombinant E2 subunit vaccine in a field setting, as a potentially safer alternative to currently used modified live vaccines against classical swine fever (CSF). Pregnant sows and their newborn piglets were evaluated for maternal transfer of vaccine derived antibodies. Farms with naïve animals were compared to farms with immunity from LOM vaccine to evaluate influence of maternally-derived antibodies to seroconversion in piglets.

Comments: This is a valuable study that provides useful information demonstrating how the recombinant E2 vaccine performs in a field setting. In general, the manuscript is clearly written and easily understood with some areas that will require clarification.

  • Please better define the purpose of the study, such as ‘to evaluate immune response/induction of antibodies in response to vaccine under field conditions with and without pre-existing CSFV antibodies’ rather than use of the term ‘efficacy’, as protection itself was not evaluated in the study. Consider providing evidence from literature if it exists that certain levels of E2 antibodies correlates with protection to suggest the levels shown in this study would be expected to be protective.
  • It would be helpful to the reader to provide more context regarding the situation of CSF on Jeju Island, such as the prevalence of virus/disease compared to mainland.
  • Consider providing a diagrammatic timeline showing the different parts of the study with timing of blood collection, vaccination and birth. This is shown nicely in the top of the tables, but would help to see it as an overview of the entire study

Line by line comments:

18, 76: (and throughout the manuscript): In the context of animals, ‘suckling’ is a more common term than ‘breastfeeding’ which is usually used relating to humans.

29-30: please briefly explain comment ‘all animals must be vaccinated’, do you mean it is a government (or some other type of) regulation for all animals to be vaccinated on mainland but not Jeju Island?

50: please explain what you mean by ‘unintended accidental use’ of LOM

66-67: briefly describe the envelope glycoproteins. Manuscript introduces Erns and E2 without describing or explaining what these are. Why is E2 is a good vaccine target antigen? Add some further detail about the vaccine – what makes it a marker vaccine (is there something different about the recombinant E2 protein compared to native viral E2, or because can identify naïve but vaccinated animals by serologic response to E2 but not other viral proteins)? Explicitly describe how measuring Erns and E2 differentiates animals that have been exposed to virus, LOM or received maternal antibodies versus recombinant vaccine. It would also be helpful to introduce concept of maternal antibodies/interference given this is an important part of the study.

143-146: was it possible this animal was previously exposed to virus or vaccine? Please state whether any of the piglets in the subsequent evaluation were from this sow’s litter – they should be excluded given it cannot be determined if their E2 antibodies are derived from the recombinant vaccine.

162-165: the Erns data for day 21 is not shown here, and it is unclear if you are referring to the A2 sow or the B1 sow – please clarify and consider presenting the Erns data. Does this imply the animal was exposed to CSFV during this period?

183-185: clarify here to explicitly state that the piglets were selected from the litters born to the sows in table 1

219: briefly describe here or in discussion possible explanation for why piglet B2 has no E2 or NA reactivity at 60 dpv

260-262: this becomes a little complicated to interpret as serum samples offset from immunization so don’t know what maternal or vaccine induced antibody levels were at time of vaccine boost and therefore difficult to interpret whether maternal antibodies were potentially interfering with the recombinant E2 vaccine – these factors should be addressed in the discussion. Additionally, the Erns data is not shown to evaluate kinetics of the maternally derived antibodies, which would enhance ability to interpret the data. I suggest including this or consider going back to measure this from the bankded serum.

272-273: as the exact timing of maternal antibody decline and vaccines are not conclusive, it is not possible to make this conclusion. Need to rethink conclusions here, address some limitations of the study, and provide greater insight in discussion

290-316: Discussion- recommend rethinking and rewriting this section. It doesn’t have a logical flow or provide insight/discussion into what is interesting and meaningful from your results and the implications and limitations of your findings.

304-307: health/vaccine status of the herds in this study should have been described in methods section, not first introduced in the discussion

307-308: this study did not specifically demonstrate there is no cross-interference, you could reword this to suggest that equivalent antibody levels in the different herds suggests no cross-interference

308-309: more detailed discussion of the role of maternal antibodies would be worthwhile – the manuscript looks at this concept without fully explaining or discussing it

311-312: ‘was expressed well without problems even in Erns-positive pigs’

Author Response

Dear reviewer 3,

We attached responses to your comments.

Reviewer 4 Report

The manuscript entitled: “Field application reports of plant-produced recombinant E2 green marker vaccinations of E2-negative pigs on Jeju Island and LOM-vaccinated pigs in the suburb of Pohang, Korea” Park et al., describes the efficacy of a new CSF vaccine based on plant-produced recombinant E2 protein at two different locations in Korea. The results evidenced an increase of immune response in naïve animals, pregnant sows, and young piglets. Passive immunity from mother sows to newborn piglets via breastfeeding was detected. Moreover, this vaccine showed efficacy in newborn piglets with maternal E2 antibodies transmitted from MLV-LOM vaccine-immunized mother sows.

I think this manuscript can be worth publishing if the following points are inserted:

General comments

1) Please, insert the hypothesis of the study in the Introduction section;

2) Please, indicate if an Ethics Committee has authorized the experiments;

3) Please, insert the innocuity section of plant-produced recombinant E2 green marker vaccine in pigs;

4) Please, insert the statistical analysis section used for choosing the number of animals used in the experiments;

5) Please, insert the statistical analysis section for the serological results;

6) Please, insert a Table describing the groups, the number of animals, doses and vaccinations performed.

7) Please, note that the Discussion section is too short. I suggest explaining in this section the results obtained in the study compared with other reports against CSF and other plant production of recombinant protein.   

Specific comments

1) Materials and Methods section (page 3, line 98): Please, insert the “Erns – specific antibodies were detected only on day 70 (0 dpv)”  in accordance with results described in Table 1;

2) Materials and Methods section (page 3, line 98): Please, change “Physical conditions”  into  “Clinical symptoms”;

3) Materials and Methods section (page 3, line 105): Please, change “Physical conditions” into “Clinical symptoms”;

4) Results section (page 3, lines 132-138): Please, delete this paragraph because already present in the Introduction and Materials and Methods sections.

5) Results section (page 3, lines 140-143): Please, delete this paragraph “Moreover… HERBAVACTM)” because already present in the Materials and Methods section;

6) Results section (page 3, lines 146-147): Please, clarify “The one outlier for E2-specific antibody ELISA test was not clearly understood”. I suggest you insert that the authors hypothesize “a non-specific reaction of the ELISA test used”.

7) Results section (pages 3-4, lines 147-161): Please, delete this paragraph “After confirmation… delivery)” because already present in  the Materials and Methods section;

8) Results section (page 4, lines 161-166): Please, this paragraph (Sera obtained from six… NA titer) is unclear and confusing. I suggest rewriting the paragraph according to what is described in Table 1.

9) Results section (page 4, line 169): Please, clarify the term SD (standard deviation?).

10) Results section (pages 4-5, lines 182-185): Please, transfer this paragraph “To gain… NA titers)” to the Materials and Methods section;

11) Results section (page 5, line 187): Please, change the value 96.35 to the higher value of 96,58 reported in Table 2 column E2.

12) Results section (page 5, lines 203-207) Please, delete this paragraph “Next,… Na titers” because already present in  the Materials and Methods section;

13) Results section (pages 5-6, lines 213-215) Please, delete this paragraph “To further… assay)” because already present in  the Materials and Methods section;

14) Results section (page 9, lines 234-242) Please, transfer this paragraph “In this experimental… Na titers” to the Material and Methods section:

15) Results section (page 9, lines 249-252) Please, delete this paragraph “The first immunization… Na titers)” because already present in  the Materials and Methods section;

16) Results section (pages 9, lines 264-266) Please, delete this paragraph “Nex, animals… Na titers)” because already present in  the Materials and Methods section;

Author Response

(The authors gave the same response as above.)

Round 2

Reviewer 3 Report

Authors have significantly improved the manuscript.

Minor editing of the changes (to address a few spelling/grammatical errors) will be required but the content and presentation are ready for publication.

Reviewer 4 Report

Accept in present form.